# LEARNING PROGRESS-GUIDED LLM GOAL GENERATION FOR AUTOTELIC SKILL LEARNING

## ABSTRACT

Reinforcement learning agents typically operate within fixed goal spaces, which limits the breadth of skills they can acquire. Large language models promise to overcome this constraint through dynamic goal generation, yet prompting them for merely interesting goals rarely produces effective curricula. We evaluate open-ended curricula using two key dimensions — *learnability* and *diversity* — and show that competence-based LLM approaches generate goals that appear promising but drive limited genuine learning progress. Our method instead optimizes goal generation directly for learning progress and consistently outperforms competence-based baselines on both learnability and diversity. In the *Crafter* domain, this leads agents to acquire diverse, challenging, and practically useful skills in the absence of extrinsic rewards.

## 1 INTRODUCTION

A central ambition in AI is to create agents capable of open-ended learning — continually expanding their behavioral repertoire without externally provided tasks or rewards (Turing, 2021; Sigaud et al., 2023). Autotelic learning offers a promising path toward this goal: allowing agents to represent and generate their own goals, learn to achieve them, and through this process uncover increasingly sophisticated behaviors (Colas et al., 2022). Yet most existing approaches still rely on predefined goal spaces, ultimately limiting the diversity and novelty of what agents can learn (Portelas et al., 2020; Liu et al., 2022; Colas et al., 2022). Recently, large language models (LLMs) have offered a way forward: by generating executable code that defines reward functions, they can propose goals drawn from an effectively unbounded space of programmatic objectives (Ma et al., 2023; Faldor et al., 2024). The challenge is to ensure that these LLM-generated goals truly maximize the agents learning potential.

Existing goal-generation methods often use competence-based heuristics to target just-right difficultytasks where the agents current performance is intermediate (Florensa et al., 2018; Racaniere et al., 2019; Faldor et al., 2024). The idea is that such tasks should be neither too easy nor too hard, but this assumption conflates *competence* (current performance level) with *learnability* (potential for further improvement). Intermediate competence can arise for reasons that offer little learning opportunity: the agent might succeed only by chance, or it may have reached a competence plateau where further progress is blocked by environmental constraints or by its own limitations. In both cases, the tasks appear just-right but fail to drive genuine learning.

Empirical learning progress (LP) captures how quickly the agents competence has improved, using *past* learnability as a proxy for where further learning is likely to occur (Kaplan & Oudeyer, 2007). Earlier work exploited this signal to *select* goals from fixed spaces (e.g. Baranes & Oudeyer, 2009; Colas et al., 2019; Kanitscheider et al., 2021). We take the next step and use LP to *generate* new goals with large language models: by conditioning the LLM on contrastive historiesexamples of goals that produced high versus low LPwe bias generation toward goals that recently drove progress and are therefore better aligned with the objective of continued improvement. To keep exploration broad and prevent a collapse in the curriculum, we complement this with an automatic semantic categorization that spreads goal proposals across distinct progress niches.

Figure 1: **Open-ended curricula for autotelic agents**. The agent architecture runs in a loop with two main modules, which communicate through the goal archive. (1) A goal generator samples contrastive high- and low-learning-progress (LP) goals from an archive of past experiences and uses them to prompt a large language model to propose new, challenging goals. (2) A goal-conditioned RL module trains a shared policy on goals drawn from the archive and reports updated measures of competence and LP.

Our contributions are:

- We introduce LP-guided LLM-based goal generation: a method that conditions an LLM on contrastive high- versus low-LP goals sampled from the agent's history to bias future goal generation toward objectives that maximize future learnability.
- We propose a diversity-preserving mechanism that maintains diversity of self-generated goals across time and prevents the diversity collapse observed in methods optimizing for learning progress only.
- We present empirical validation of our proposed method on the challenging CRAFTER environment, where our LP-guided LLM-based curricula generates goals leading to more learnable, diverse, difficult and environment-aligned behaviors than competence-based baselines.

We believe our proposed metrics will foster progress in open-ended learning research and make it more accessible, while renewing interest in learning-progressbased approaches as a key driver of open-ended curriculum generation.

## 2 RELATED WORK

**Self-generated curriculum learning in goal spaces.** Curriculum learning traditionally structures training tasks to accelerate skill acquisition (Portelas et al., 2020). A specific branch focuses on self-generated curricula in goal space, where the agent autonomously chooses which goals to pursue (Colas et al., 2022). Intrinsic motivation signals have been proposed to steer this choice toward goals expected to best shape the agents learning trajectory: favoring goals in sparse areas of the goal space (Pong et al., 2019), that maximize disagreement the predictions of value networks (Zhang et al., 2020), associated with intermediate difficulty (Sukhbaatar et al., 2017; Florensa et al., 2018; Racaniere et al., 2019; Campero et al., 2020; Foster et al., 2025), or associated with recent learning progress (LP) (Baranes & Oudeyer, 2009; 2013; Blaes et al., 2019; Colas et al., 2019; Akakzia et al., 2020; Kanitscheider et al., 2021; Gaven et al., 2025). Early work also explored goal generation rather than mere selection — for example GoalGAN (Florensa et al., 2018), SetterSolver (Racaniere et al., 2019), or the adversarial self-play of Sukhbaatar et al. (2017) — but these efforts were made in low-dimensional, hand-engineered goal spaces, limiting their potential for truly open-ended skill discovery.

**Autotelic RL for open-ended learning.** Autotelic learning extends these ideas beyond goal selection: agents learn goal representations and generate their own goals to sustain open-ended skill acquisition (Colas et al., 2022; Sigaud et al., 2023). Approaches include training autoencoders on past states (Péré et al., 2018; Nair et al., 2018; Cully, 2019), discovering maximally discriminative skills or goals (e.g. Eysenbach et al., 2018), or internalizing reward functions from linguistic feedback (Colas et al., 2020). A limitation of these methods is that the representation of goals must itself be learned and may have to evolve as the agents competence grows. Foundation models change this: they provide *universal representational spaces* in which goals can be expressed from the outset, eliminating the need for the agent to continually adapt its goal language. Examples include linguis-

tic goals whose achievement is judged by visionlanguage models or captioners (Du et al., 2023b;a), and programmatic goals whose achievement is assessed by executing reward-producing programs (Ma et al., 2023; Faldor et al., 2024; Zhao et al., 2025; Chen et al., 2025).

**Goals as reward programs.** Large language models now make it practical to generate such reward programs automatically: Eureka (Ma et al., 2023) uses LLMs to write reward functions for robotic skill learning, while OMNI-EPIC (Faldor et al., 2024) goes further by generating entire environments for open-ended reinforcement learning. More recently, work on strengthening the reasoning abilities of LLMs has started to leverage automatic task generation in domains such as coding and mathematics (Pourcel et al., 2024; Zhao et al., 2025; Chen et al., 2025). Our approach is closest to these efforts in using LLMs to create programmatic goals for open-ended learning, but it differs in two key ways: we introduce a metric suite to evaluate the open-endedness of the resulting curricula, and we steer goal generation by *learning progress* rather than by competence-based heuristics.

## 3 METHOD

### 3.1 PROBLEM DEFINITION

We conduct our study in a goal-augmented MDP $(\mathcal{S}, \mathcal{A}, \mathcal{T}, \mathcal{G}, \mathcal{R}, \gamma)$, with $\mathcal{S}$ a set of states, $\mathcal{T}$ the transition function, $\mathcal{A}$ the action space, $G$ the goal space, $R$ the space of binary reward functions indicating whether a state $s$ satisfies a goal $g \in \mathcal{G}$ and $\gamma$ the discount factor. In our setting, a goal $g$ is a tuple $g = (nm_g, \mathcal{R}_g)$, where $nm_g \in \mathcal{V}^{L_{nm_g}}$ is the name of the goal, $\mathcal{V}$ the vocabulary and $\mathcal{R}_g \in \mathcal{R} \subset \mathcal{V}^{L_\mathcal{R}}$ is the associated reward function in the form of a code. Here $(L_{nm_g}, L_\mathcal{R}) \in \mathbb{N}^2$ are respectively the maximum length of the goal name and maximum length of the code for the reward function corresponding to the goal. The agent is modeled by a goal condition policy $\pi : \mathcal{S} \times \mathcal{G} \to \mathcal{A}$. The objective of our method is to design a goal generator function $\mathfrak{G}$ that is used to efficiently train an agent to master a variety of tasks.

To do so, we want that a goal $g$ generated by $\mathfrak{G}$ maximizes two metrics, the learning progress $LP$ and the distance $d_{\text{emb}}$ between $g$ and all previously generated goals. We define these metric as:

$$LP(g, k_{init}, k_{end}) = \max_{k \in [|k_{init}, k_{end}|]} (SR(k, g) - SR(k_{init}, g)),$$

where $k_{init}$ is the step at which the agent has trained on $g$ for the first time and $SR(k, g)$ is the success rate for goal $g$ at step $k$,

$$d_{\text{emb}}(g_i, g_j) = ||\mathcal{E}(n_{g_i}) - \mathcal{E}(n_{g_j})||_2,$$

where $\mathcal{E} : \mathcal{V}^{L_{nm_g}} \to \mathbb{R}^{n_e}$ an embedding function that map the semantic description $nm_g$ of a goal $g$ to a vector in $\mathbb{R}^{n_e}$.

We condition $\mathfrak{G}$ on the history of goals already generated:

$$\mathcal{H}_{ist} = \{(g, SR(g)), \forall g \in \mathcal{G}_{att}\},$$

where $\mathcal{G}_{att} \subseteq \mathcal{G}$ the set of goals attempted by the agent.

### 3.2 METHOD OVERVIEW

The agent operates through two distinct modules (Fig.1) to facilitate the open-ended generation and learning of goals. In the first module, an LLM model $\mathfrak{G}$ adaptively generates goals based on the agent's current skill level. All tasks are stored in an archive $\Lambda$ that approximates $\mathcal{H}_{ist}$. In the second module, a goal-conditioned deep RL agent learns the tasks from the archive.

The process is initialized by adding a set of hand-crafted goals to the archive. The manually created goals are very simple and allow the LLM to learn how to generate syntactically correct reward functions. The multi-goal agent is then trained on these goals for a few updates, involving data collection steps and weight updates. Then, based on the success of the agent in learning the different tasks, the archive is updated: tasks that are too difficult or too old are removed from the archive. This iterative process allows the model to continuously generate and adapt goals that align with the agent's evolving capabilities, fostering an open-ended learning environment. In the next two sections, we first present how the goal generator is modeled by an LLM, then how goal condition policy is trained on the generated goals. Figure 1 illustrates the process described above and the corresponding pseudo-algorithm is given in Figure 3.

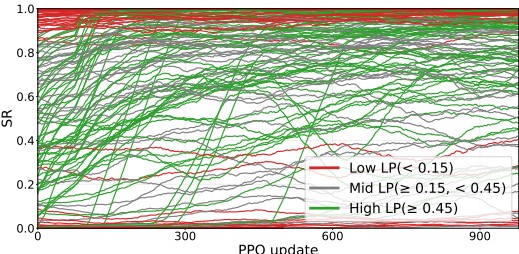

Figure 2: **Learning curves over two consecutive curriculum iterations**. Green and red indicate the sets of high-LP (positive) and low-LP (negative) examples the agent samples from to elicit novel high learnability goals from the LLM goal generator. Low-LP goals can be either too easy (bottom red lines), too hard (top red lines) or sometimes intermediately difficult but without triggering any learning (middle red lines). The gray lines represent goals for which the signal is not clear enough to be used as an example.

### 3.3 THE GOAL GENERATOR

$\mathcal{H}_{ist}$ can become too big to be used in practice, thus we approximate it with $\Lambda$. This set contains at most $n_\Lambda$ tuples of the form $(nm_g, \mathcal{E}(nm_g), \mathcal{R}_g, SR(g), LP(g))$. We model the goal generator using an LLM, which is prompted with in-context examples of goals separated in three groups based on their current LP and SR. Thus, we have the positive examples with high LP, the negative hard examples with low LP and low SR and the negative easy examples with low LP and high SR. These examples inform the LLM about the capacities of the goal-conditioned agent and are selected in $\Lambda$. Figure 2 shows the evolution of the agent success rate for several goals and how they are classified into the different categories; in particular, we do not use goals for which the LP is neither high nor low, as they do not give a clear enough signal as in-context examples. In order to force diversity, we first clustered $\Lambda$ into different categories of goals generated by the LLM (see Appendix D). Then, for each curriculum iteration, select the in-context examples inside one of these categories. Thus, the goal generator proposes new goals or modifies existing ones to maximize the estimated $LP$ (the tasks, although difficult, must be learnable).

After generation, a procedure (see Appendix A.3) removes goals $g$ whose reward functions $\mathcal{R}$ are not syntactically correct (that do not compile). $\mathcal{R}$ can use privileged information such as the current state of the game engine, the agent's current action, and a reward state where memory can be stored. This allows the agent to target challenging goals such as time-extended goals (e.g., "move up three times") and goals involving optimization under selected constraints (e.g., "build a shelter while maintaining your health above 5"). Examples of such reward code are provided in Appendix C. The generated goals are then added to $\Lambda$.

After each curriculum iteration, all goals in the archive from the previous iteration are ranked and $n_{worst}$ are removed from $\Lambda$. To classify the goals, we use a fitness function $f(g) = \text{iteration}(g) \times SR(g)$, where iteration$(g)$ is the iteration at which the goals have been generated $g$.

### 3.4 THE GOAL CONDITIONED LEARNER

While any RL algorithm could be used, we train the goal-conditioned agent on goals in $\Lambda$ using PPO. Appendix B.1 details the hyperparameters used for this training. The agent is a causal transformer (Dai et al., 2019), conditioned on both textual and visual information. The textual information is the embedding of the $nm_g$, $\mathcal{E}(nm_g) = z \in \mathbb{R}^{n_e}$, with $n_e$ being the dimensionality of the embedding space. The visual information consists of the last $n_{obs}$ images returned by the environment. The goals are sampled in the archive $\Lambda$ following a uniform distribution. At the end of each trajectory, the LP and SR of a goal are updated.

Figure 3 shows the pseudocode of the pipeline used in the experiments.

```python
# archive with 36 hand-crafted goals, archive capacity is 200
goal_archive = [goal_1, goal_2, ....., goal_36]

# randomly initialized goal-conditioned RL agent
agent = create_rl_agent()

# an LLM object with implemented functions to cluster and generate goals
LLM = create_llm()

# start main loop
for iteration in 8:
    # 1. Goal-conditioned RL training
    for update in 500:
        goals = sample_goals(200)
        trajectories = agent.rollout(goals, 200*100)
        success_rates = compute_sr(goals, trajectories)
        lps = compute_lps(goals, trajectories)
        agent.train(trajectories)

    goal_archive.update_metrics(goals, success_rates, lps)

    # 2. Category generation and annotation

    # create categories based on 85 goals
    sample_goals = goal_archive.random_sample(85)
    sample_categories = LLM.cluster_goals(sample_goals, n_categories=5)

    # annotate the remaining goals using the generated categories
    archive.goal.categories = LLM.annotate_goals(
        archive.goals,
        sample_goals, sample_categories
    )

    # 3. Goal generation
    # generate 115 new goals
    new_goals=[]
    while len(new_goals) < 115:
        category = sample(set(categories), 1) # sample a category

        # Sample in-context examples
        # for LP:
        #    positive (lp>0.45), negative_easy (lp<0.15, sr>0.5),
        #    and negative_hard (lp<0.15 sr<0.5)
        # for OMNI-EPIC:
        #    positive (sr>0) and negative (sr=0), following Faldor et al. (2024)
        # for uniform:
        #    positive only
        in_context_examples = archive.sample_in_context_examples(category, 6)

        if len(in_context_examples.positive_examples) == 0:
            continue

        new_goal = llm.generate_goal(in_context_examples, category)

        # check goal validity (see Appendix A.3 for details)
        if goal_is_valid(new_goal)
            new_goals.append(new_goal)

    # 4. Archive filtering
    # first remove goals with sr=0, then remove oldest goals until a total of 115 removed
    zero_sr_goals = archive.extract_goals_with_sr_zero()
    archive.remove_goals(zero_sr_goals)
    n_zero_sr_goals_removed = len(zero_sr_goals)

    if n_zero_sr_goals_removed < 115:
        to_remove = 115 - n_zero_sr_goals_removed

        archive.remove_oldest_goals(to_remove)
```

Figure 3: **Pseudocode of the autotelic agent**: at each curriculum iteration the agent is trained on the goals of the archive (Section 3.4), then new goals are generated and the archive is updated (Section 3.3).

## 4 EVALUATION METRICS

We evaluate the quality of a curriculum along two main dimensions: *learnability* and *diversity*.

**Learnability**  At iteration $t$, the ongoing learnability of the curriculum is the total competence gain the agent has achieved on the archived goals: $\sum_{g \in \text{archive}} SR^g_{\max}$ - $SR^g_{\text{start}}$, where $SR^g$ is the success rate of the agent on goal $g$ computed from a windows of 1.25M environment steps with uniform goal sampling. $SR^g_{\text{start}}$ is computed over the initial window, while $SR^g$max is the maximum $SR$ found over the current curriculum iteration. The overall learnability of the curriculum is then the cumulative sum of the iteration-specific learnabilities, capturing the total competence progress across all goals over training.

**Diversity**  We compute the diversity of a given iteration of the curriculum as the pairwise L2 distance between the embeddings of all goals generated at that iteration (computed with "text-embedding-3-small"). The final diversity of the curriculum is computed the same way on the total set of all generated goals.

To make sure generated goals drive the learning of useful skills, we track three additional metrics:

**Relative difficulty**  We estimate the relative difficulty of a goal by measuring the area between the learning curve of the agent with the one of a randomly initialized agent, each given 500 training updates. We add this measure for all goals generated at the current iteration, and all goals generated over the whole learning trajectory to obtain the iteration-specific and final measures respectively.

**Environment alignment**  This metric measures the extent to which the curriculum led the agent to acquire skills that are useful in its environment. In Crafter, we use the *Crafter score*, a metric designed to increase as the agent unlocks more of the environment achievements (e.g. collecting a drink, defeating a zombie, making a stone sword), see full list in Appendix A.1. It is computed as $score_t = exp(1/K \sum_{k=1}^{K} log(a_k + 1)) - 1$, where $a_{k,t}$ is the ratio of episodes unlocking achievement $k$ at iteration $t$.

**Interestingness**  We estimate interestingness using an LLM-as-a-judge setup following Zhang et al. (2024) where an LLM was used a as proxy of the "human notion of interestingness". For each baseline 200 random goals with $LP > 0.45$ are sampled and given to the "gemini-2.0-flash" model to judge the interestingness from 0 to 10, see prompt in Appendix A.2.

## 5 EXPERIMENTS

In the section we aim to address the following scientific questions:

- Does learning progress foster better goal generation?
- Does the introduces category construction mechanism help prevent diversity collapse?
- Are the discovered goals meaningful with respect to the environment?
- What kind of categories and goals are discovered by the LP-based curriculum?

### 5.1 DOES LEARNING PROGRESS FOSTER BETTER GOAL GENERATION?

In this experiment we aim to evaluate the benefit of LP for goal-generation based exploration. We compare the performance of using LP to sample in-context examples to two other approaches "OMNI" (which focuses on success rates following Faldor et al. (2024)) and "Uniform" (which samples the examples uniformly). To clarify, all of the three approaches also use the category construction mechanism proposed in 3.

Figure 4 compares aforementioned approaches in terms of Learnability, and Diversity (Mean $\pm$ SEM, 3 seeds). We can see that "LP" confidently outperforms other baselines in terms of Learnability indicating that our curriculum was able to discover more learnable tasks. Regarding diversity, while "LP" does not outperform other baselines but it is interesting to note that "OMNI" exhibits

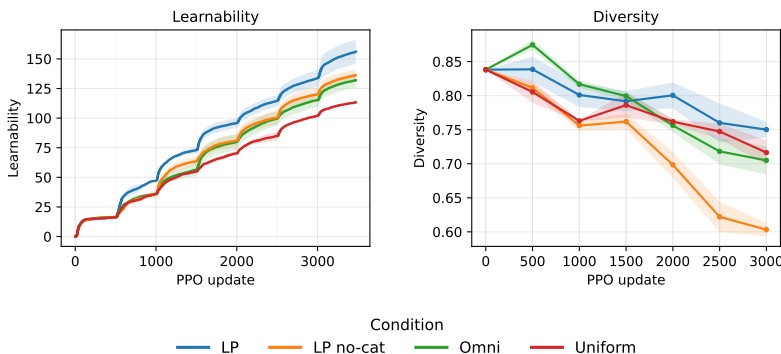

Figure 4: **Learnability and Diversity comparison of the proposed approach**. We compare our full approach "LP", which uses both LP and category construction, to "OMNI" and "Uniform", which use category construction and their respective in-context sampling methods. We can see that both introduced mechanisms - LP and category construction - improve exploration. LP leads to higher learnability, and category construction prevents a drastic loss in diversity.

a higher diversity in the first iteration which then quickly drops, and is then superseded by "LP" towards the end (which appears more stable). Overall, this experiment shows the benefit of using the introduced LP-based goal generation mechanism as it leads to goals of higher learnability.

## 5.2 DOES THE CATEGORY CONSTRUCTION MECHANISM MITIGATE DIVERSITY COLLAPSE?

We next test the impact of the categoryconstruction mechanism described in Section 3. Because of computational limits we focus on the best-performing variant from the previous experiment (LP). The ablation LP_no_cat removes the category mechanism.

Figure 4 reports the diversity of generated goals over curriculum iterations. Without categories, diversity in LP_no_cat drops sharply, whereas the full LP method maintains a broad set of goals. The mechanism works by partitioning discovered behaviors into semantic *niches* and explicitly sampling goals from each niche; this spreads proposals across distinct behavior families and seeks learning progress within every niche.

Interestingly, adding categories does not reduce *learnability* but increases it. Although the sampler no longer concentrates exclusively on the single niche with the highest immediate learning progress, the agent still achieves equal or better competence gains. This suggests efficient transfer: exploring a wider variety of niches uncovers areas that later yield high progress, compensating for the temporary dilution of focus. Overall, category construction prevents drastic diversity collapse while still discovering highly learnable goals, highlighting its value for sustaining open-ended skill growth.

## 5.3 ARE THE DISCOVERED GOALS MEANINGFUL WITH RESPECT TO THE ENVIRONMENT?

We next assess whether the curriculums generated goals lead the agent to acquire behaviors that matter in the environment. We track three complementary metrics (Section 4): *environment alignment* — the extent to which learned behaviors exploit the environments affordances; *relative difficulty* — how hard those behaviors would be for an agent trained from scratch; and *interestingness* — whether humans would find the behaviors noteworthy.

Figure 5 summarizes the results. For all methods, the Crafter score rises steadily across curriculum iterations, reaching about 4–5, comparable to the best scores reported for autotelic approaches that rely on textual observations and custom captioners (Du et al., 2023b). This indicates that LLM-based goal generators can adapt their proposals to the environments specific affordances. Relative difficulty also increases in the early stages, showing that the curriculum progressively challenges the agent with goals that would be harder for a nave agent, effectively scaffolding its learning trajectory.

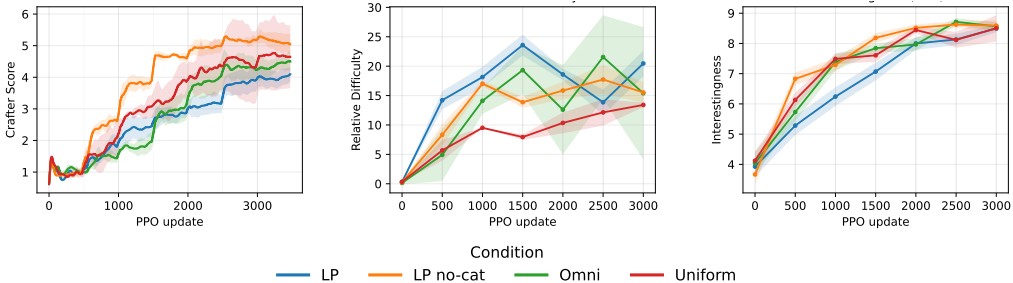

Figure 5: **Measuring curriculum relevance**. Crafter score, relative difficulty, and interestingness all rise across curriculum iterations, showing that LLM-generated goals drive agents toward behaviors that are environmentally meaningful and human-interesting. The trends are similar for LP-, competence-, and uniform-based example selection, indicating that these qualities stem primarily from the shared LLM goal generator rather than the specific prompting strategy.

Interestingness follows a similar upward trend: as the curriculum proposes harder goals, these are judged more compelling by the LLM evaluator.

Taken together, these three metrics confirm that LLM-generated curricula drive agents toward behaviors that are both environmentally relevant and human-interesting. However, we observe no significant differences between LP-based, competence-based, or uniform in-context selection. This suggests that these qualities are largely determined by the shared LLM generator rather than by the specific example-selection strategy. Future work could investigate explicit optimization mechanisms to further enhance environment alignment, relative difficulty, and interestingness.

### 5.4 WHAT KIND OF CATEGORIES AND GOALS ARE DISCOVERED BY THE LP-BASED CURRICULUM?

Here we qualitatively explore the progression of goals and categories discovered by our approach. We take all goals generated in one run of the approach that uses both LP and category construction. We fit a T-SNE model on the goal embeddings (constructed as described in Section 3.1). We additionally annotate the goals from the first and last iteration with the categories to which they were assigned. Figure 6 shows the latent representation of generated goals. We can see that goals from the earlier generations, such as those corresponding to basic navigation, exploration, resource acquisition and simple manipulation and placement of object, are predominately placed in the bottom right corner. Similarly, goals from the later generations, such as those corresponding to resource acquisition, navigation, compositional goals, complex environmental interactions and construction of simple structures, are predominantly placed in the middle and top left of the latent space. Having said that, there are some goals from the later iterations in the bottom right corner, as well as some goals from the earlier iterations in the top left corner. This means that the curriculum maintains some smaller proportion of simpler goals, which is beneficial as it enables to maintain performance of simpler tasks as well. Overall, this experiment shows that the curriculum gradually moves from simpler to more complex goals.

## 6 DISCUSSION

We introduced a framework for *learning-progress–guided* goal generation in open-ended reinforcement learning. By conditioning a large language model on contrastive examples of high- versus low-LP goals, we bias goal proposals toward objectives with the greatest potential for future improvement. In the CRAFTER environment this approach (i) yields consistently higher *learnability* than competence-based or uniform baselines, (ii) maintains goal *diversity* through the proposed category-construction mechanism, and (iii) produces goals whose difficulty, environment alignment, and human-judged interestingness increase throughout training. These results show that learning progress is an effective intrinsic signal for steering LLM-driven curriculum generation, enabling

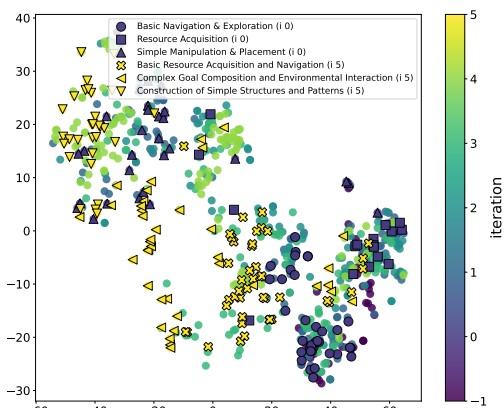

Figure 6: **T-SNE representation of discovered goals and categories**. We can see that in the beginning the goals are more focused on simpler tasks such as simple navigation and exploration and in the end on tasks relating to more complex environment interaction, compositional goals and construction of structure. Furthermore, we can see that the curriculum gradually moves from simpler to more complex goals.

agents to acquire a broader and more genuinely learnable repertoire of skills without external rewards.

Beyond higher scores, these results show that *how* goals are proposed shapes the long-term dynamics of open-ended learning. LP-guided prompting ties goal generation to a measurable signal of continued progress, steering exploration without hand-crafted curricula. Category construction acts as a simple but effective form of open-ended exploration pressure: by forcing the generator to cover multiple semantic niches it prevents early specialization and uncovers future regions of progress. Together these mechanisms illustrate a scalable way to couple large language models with intrinsic-motivation signals, turning the LLMs universal reward-program space into a practical substrate for sustained skill growth.

Our study is limited in several aspects. First, all experiments were performed in a single simulated domain (CRAFTER); demonstrating that LP-guided goal generation scales to other environments or to realworld robotics remains future work. Second, environment alignment, relative difficulty, and interestingness were largely unaffected by the in-context sampling strategy, indicating that these aspects are currently dominated by the shared LLM generator rather than by our prompting method. Finally, the computational budget constrained the size of the goal archive and the frequency of evaluation; richer or largerscale settings may reveal additional challenges or require more sophisticated filtering and evaluation procedures.

Several extensions naturally follow. A first step is to couple LP with additional objectivessuch as explicit optimization for environment alignment, relative difficulty, or human-interestingnessto guide the generator toward goals that are not only learnable but also societally relevant. Adaptive or hierarchical category construction could let the niches evolve as new behaviors emerge, sustaining diversity without manual tuning.

This work demonstrates that combining learning progress with large language model goal generation provides a promising route to sustained open-ended skill acquisition.

ETHICS STATEMENT

The project does not involve any ethically concerning aspects. No datasets were used or created and the experiments are in a simulated artificial environment which does not afford the expression of socially damaging biases. The scientific and engineering contributions presented here are very general and as such could be applied to many different usecases. Future applications of this methods should maintain caution so as to not apply it for socially damaging or unethical aims.

## REPRODUCIBILITY STATEMENT

Many technical details are provided in the main text in Section 3, and in the Appendix A. Hyperparameters used for the whole pipeline and the RL training are provided in the Appendix B and prompts in Appendix D. We will fully open source the code, which will enable to easily recreate and extend our experiments. Furthermore, all experiments were conducted with three seeds and we depicts standard error interval for all results. This diminishes the problems of stochasticity in our results.

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
