# OpenReview forum: "Learning Progress-Guided LLM Goal Generation for Autotelic Skill Learning"
_ICLR.cc/2026/Conference — Submitted to ICLR 2026_

### Official Review · Reviewer_B8YZ · 2025-10-29

**Soundness:** 3
**Presentation:** 3
**Contribution:** 3
**Rating:** 2
**Confidence:** 4

**Summary:**

This paper introduces a method for autotelic reinforcement learning where goal generation by a large language model (LLM) is guided by learning progress (LP). The approach uses contrastive examples—sampling both high- and low-LP goals from the agent’s history—to prompt the LLM to propose new goals that are more likely to foster learning. A semantic clustering mechanism is also incorporated to help maintain goal diversity over time. Evaluated in the Crafter environment, the method outperforms competence-based and uniform sampling baselines in terms of learnability and diversity.

**Strengths:**

Well-Structured Pipeline: The authors present a complete and systematic framework that covers the full loop of goal generation, semantic clustering, archive management, and goal-conditioned policy learning. The inclusion of a diversity-preserving mechanism effectively mitigates the risk of goal space collapse.

High Reproducibility: The paper offers detailed algorithmic pseudocode and hyperparameters which will significantly facilitate replication and future extension of their work.

**Weaknesses:**

1. Limited Experimental Validation: The empirical evaluation is conducted exclusively within the *Crafter* environment. While complex, this single-domain validation severely limits the claims of open-endedness and generalizability.

2.  Limited Novelty and Algorithmic Contribution: The core idea of using Learning Progress (LP) as an intrinsic signal is not novel, as there is a substantial body of prior work using LP for goal *selection* (as thoroughly covered in the related work). The contribution of this paper appears to be the *integration* of this classic intrinsic motivation signal with a modern LLM-based generator. The paper does not sufficiently demonstrate why this specific combination yields a qualitative leap over existing methods, beyond what might be expected from a powerful LLM generator being guided by any reasonable signal.

3.  Unclear and Inconsistent Notation: The problem formulation and methodology are unclear and inconsistent, which obscures the core algorithm.
*   In the `d_emb` formula, it is written as `E(n_gi)`, whereas in the preceding goal definition, the name is `nm_g`. Is `n_gi` a typo for `nm_gi`, or does it refer to something else?
*   Is `SR(g)` a static value? This seems to contradict the previously defined `SR(k, g)`, which varies with step `k`. Is the value stored in the history the most recent success rate, the average success rate, or the success rate at a specific point in time? This definition is crucial for understanding how high/low LP examples are selected from the history.

**Questions:**

see weaknesses

---

### Official Review · Reviewer_jACY · 2025-10-31

**Soundness:** 3
**Presentation:** 2
**Contribution:** 2
**Rating:** 6
**Confidence:** 2

**Summary:**

A challenge for RL in open-ended environments is self-generating goals. The authors propose that, rather than generate goals based on expected difficulty (assumed to then help drive learning progress), it's better to learn to generate goals that directly optimize for learning progress. Their evaluation substantiates this claim.

**Strengths:**

- The basic premise of the paper seems sound, and is a creative idea to explore.
 - The evaluation substantiates the claim convincingly
 - As far as I can tell the formalism is clean and sound

**Weaknesses:**

- The paper is frustratingly short of examples, making it hard to read and mentally ground. In particular, the Crafter domain is not described anywhere that I can find beyond a single hint in the appendix that it's a kind of Minecraft-game-like setup. Adding in examples, e.g., to illustrate more concretely the differences between competence-based goal setting and LP-based goal settings, would significantly strengthen the paper. Some of this can be done in the appendix if you are short of space, but illustrations of some of your key points in the paper body itself would help.
 - Evaluation on a 2nd domain would add strength to the results (although the single domain results are convincing enough for me)

Minor:
 - Figure 2 is unhelpful and it's unclear how it conveys the main points in the caption. Is there a different way of presenting this?

**Questions:**

See weaknesses

---

### Official Review · Reviewer_NzCJ · 2025-10-31

**Soundness:** 2
**Presentation:** 1
**Contribution:** 2
**Rating:** 2
**Confidence:** 3

**Summary:**

The paper proposes an LP-guided LLM-based goal generation method for open-ended reinforcement learning. It uses a large language model to generate new goal programs by conditioning on examples of goals with high and low learning progress from the agent's past experience. A category construction mechanism groups goals into diverse clusters to prevent collapse toward similar objectives. The authors introduce a metric suite including learnability, diversity, difficulty, interestingness, and Crafter score---to evaluate open-endedness.

**Strengths:**

* The paper explores an interesting direction using LLMs for open-ended goal generation in reinforcement learning.

**Weaknesses:**

* Frequent grammar and spelling mistakes make the paper difficult to read
* Poor sentence structure and inconsistent phrasing reduce clarity (especially, in introduction
* Only one environment (Crafter) is used for all experiments
* It seems the novelty is limited and the paper states that its main distinction from prior work is the use of learning progress instead of competence and the introduction of an evaluation metric suite. However, it appears that learnability, the main reported metric, is directly aligned with the optimization signal (learning progress), making the evaluation somewhat circular and less informative about real generalization.

**Questions:**

* Why was Crafter chosen as the sole environment?
* Do they expect the approach to generalize beyond Crafter, and what limitations might arise?
* Was any manual or quantitative verification done to check that LLM-generated categories remain consistent across runs?
* The paper would be clearer if the authors explicitly listed which LLMs were used for each component. The interestingness score is produced by another LLM (gemini-2.0-flash). Was any human validation or manual spot check done to confirm that these ratings reflect behaviors that humans would actually consider interesting, and not just what the judge model prefers?
* Relative difficulty is defined as the area between the learning curves of a trained agent and a randomly initialized agent. Can the authors justify why that area is a stable measure of difficulty?
* Have you quantified how often the same goal is assigned to different categories across runs? Even a small agreement score (e.g.,% match) would help establish robustness.
* In LP_no_cat, is the only change the removal of category construction, with all other settings (prompting, LP thresholds, archive size, rollout budget) kept identical? Clarifying this would help isolate what the ablation is really testing.
* The paper claims the agent acquires "diverse, challenging, and practically useful skills," but the main text does not include qualitative rollouts or trajectory examples of those skills.
* It would make the paper clearer if the authors could correct grammar and phrasing in several core sections (Introduction, Contributions, Evaluation Metrics). Currently many sentences are hard to parse (e.g. "just-right difficultytasks"), which makes it harder to assess the actual contribution.

---

### Official Review · Reviewer_TwBW · 2025-11-11

**Soundness:** 3
**Presentation:** 2
**Contribution:** 2
**Rating:** 4
**Confidence:** 2

**Summary:**

This paper shows that competence-based LLM approaches generate goals that appear promising but drive limited genuine learning progress. The proposed method instead optimizes goal generation directly for learning progress and consistently outperforms
competence-based baselines on both learnability and diversity.

**Strengths:**

- The topic is important in several aspects.
- The results seem interesting and effective.

**Weaknesses:**

- The paper is hard to understand. Lack of examples makes it harder to pass.
- How to categorize the learning process vs the outcome that drives the goal generation? Also, why do we expect the behavior to be different in these two cases?

**Questions:**

See weakness.

---

### Meta-Review · Area_Chair_Cvi9 · 2026-01-05

**Summary:**

The paper posits that LLM struggle to create effective curricula when the goals they are required to generate are based solely on a notion of interestingness. The authors instead propose generating goals through a measure of learnability and diversity. They evaluate their method on the open-ended Crafter domain.

One of the main concerns of the reviewers was that Crafter is a single domain, which may limit the validity of the results. However, Crafter is an open-ended environment that contains a widespread of complex situations which are far from trivial. Therefore, I do not take into account this criticism in my meta review. However, I do agree with the reviewers that the authors could have done a better job at presenting the environment itself, as it is not as widely known as math or code benchmarks in the LLM world.

One of the main issues with the paper is presentation. It hinders reading to some extend, but to a greater extent the results themselves are far from convincing. Figures like Figure 2 are simply confusing and don't convey any meaningful idea, whereas Figure 3 are too long and lack a concise description of the important bits. Figure 5 itself shows very little upside (if any at all) to the method, which directly clashes with the main claim of the authors, that learnability-based goal generation is significantly better than competence based. On this point, there is not even a clear presentation of the baselines that rely on competence to generate goals.

Additionally, the paper could be improved by being better grounded in the literature of learning progress and diversity. These measures are far from trivial, and the current instantiations have their drawbacks, which are not clearly investigated. For example, what is the impact of the embedding function for measuring diversity?

**Reviewer Concerns:**

No concerns addressed.

**Reviewer Scores:**

Reviewer TwBW : Unlikely given the very short review.
Reviewer NzCJ: Likely, given that most of the concerns seem to be related to grammar and typos.
Reviewer jACY: Likely, the reviewer asked for a better presentation of the method, environment and results, which are fair points.
Reviewer B8YZ: Unlikely, since the reviewer presented fundamental flaws that would be hard to revert in a rebuttal.

---

### Decision · Program_Chairs · 2026-01-26

Reject